Quiz-style online training tool helps to learn birdsong identification and support citizen science

Ogawa Yui ogawa.yui.ta@alumni.tsukuba.ac.jp ogawa.yui@nies.go.jp 1 2
Fukasawa Keita 1
Yoshioka Akira 3
Kumada Nao 1
Takenaka Akio 4
Ito Taiichi 5
1 Biodiversity Division, National Institute for Environmental Studies , Tsukuba , Ibaraki , Japan
2 University of Tsukuba , Tsukuba , Ibaraki , Japan
3 Fukushima Regional Collaborative Research Center, National Institute for Environmental Studies , Miharu , Fukushima , Japan
4 Unaffiliated , Tsukuba , Ibaraki , Japan
5 National Parks Awareness Center, Edogawa University , Nagareyama , Chiba , Japan
Shrivastava Kush
Electronic publication date: 2023 May 31
Publication date: 2023
Volume: 11
Electronic Location ID: e15387
Received 2022 Nov 17; Accepted 2023 Apr 19
Copyright: ©2023 Ogawa et al.
Copyright year: 2023
Copyright holder: Ogawa et al.
License: This is an open access article distributed under the terms of the Creative Commons Attribution License, which permits unrestricted use, distribution, reproduction and adaptation in any medium and for any purpose provided that it is properly attributed. For attribution, the original author(s), title, publication source (PeerJ) and either DOI or URL of the article must be cited.
License URL: https://creativecommons.org/licenses/by/4.0/

Keywords: Citizen science, Community science, Adaptive learning, Online training, Birdsong identification, Serious game

Funding: The authors received no funding for this work.

==============================
Citizen science is an important approach to monitoring for biodiversity conservation because it allows for data acquisition or analysis on a scale that is not possible for researchers alone. In citizen science projects, the use of online training is increasing to improve such skills. However, the effectiveness of quiz-style online training, assumed to be efficient to enhance participants’ skills, has not been evaluated adequately on species identification for citizen science biodiversity monitoring projects. Memory mechanisms in adaptive learning were hypothesized to guide the development of quiz-based online training tools for learning birdsong identification and for improving interest in birds and natural environments. To examine the hypothesis, we developed a quiz-style online training tool called TORI-TORE. We experimentally applied TORI-TORE in Fukushima, Japan, and examined its effectiveness for bird identification training using test scores and questionnaires to determine participants’ attitudes in a randomized control trial. We obtained the following key results: (1) TORI-TORE had positive effects on test scores and trainees’ attitudes toward birds. (2) Adaptive training, in which questions focused preferentially on unmastered bird species based on the answer history of individual trainees inspired by adaptive learning, unexpectedly led to lower scores and satisfaction in TORI-TORE. (3) Focusing on species that are relatively easy to remember, short lag times between training and testing, and long question intervals positively affected scores. While there is room for improvement, we expect TORI-TORE to contribute to online capacity building and to increase interest in natural environments.

Introduction

For biodiversity conservation, it is necessary to monitor changes in the natural environment and ecosystems over large spatial and temporal scales. Monitoring efforts often focus on indicator species or groups, such as birds (Greenwood, 2007). Among bird monitoring methods, sound recordings are often used owing to the ability to identify species, even when they cannot be easily seen in the field (Priyadarshani, Marsland & Castro, 2018).

Citizen science, which refers to public participation in scientific research, can be used to obtain data over long periods and at large spatial scales (Cohn, 2008; Silvertown, 2009; Bonney et al., 2009a). This approach can be used to analyze sound recordings of birds (Oliver et al., 2020; Cottman-Fields, Brereton & Roe, 2013; Fukasawa et al., 2017a; Truskinger et al., 2011). Training to improve identification skills is expected to increase the data quantity and quality (Greenwood, 2007; Bonney et al., 2009b; McLaren & Cadman, 1999) and to improve the participants’ interest in the natural environment (Hsu, Chang & Liu, 2019). System for automated birdsong recognition has been developed considerably (for example, Wood et al., 2022; Jäckel et al., 2023), and a library of sounds annotated by citizens will contribute to further improving accuracy of automated birdsong recognition.

In citizen science projects, the use of online training is increasing (Bonney et al., 2009b; Gray et al., 2017), particularly during the COVID-19 pandemic (Mukhtar et al., 2020). Online training is characterized by low costs and easy accessibility (Starr et al., 2014; Hemment, Woods & Ajates Gonzalez, 2018; Ratnieks et al., 2016). However, few studies have examined the effectiveness of online training for species identification in citizen science projects (Starr et al., 2014). Quiz-style training is more efficient than simple memorization due to the testing effect, in which recalling information strengthens memory more than simply writing or listening to the information (Roediger & Karpicke, 2006). Online quizzes have been used for short-term birdsong identification training, including Bird Song Hero (Bird Academy, 2022), Photo + Sound Quiz (eBird, 2022), Larkwire (Larkwire, 2022), Bird Research Birdsong Quiz (Japan Bird Research Association, 2022). However, in existing quiz-based online training, participants repeatedly and/or randomly listen to full-length sound sources, irrespective of their levels of proficiency in identifying bird songs, or they have to customize the training content (select the set of songs) themselves. Therefore, more efficient and user-friendly quiz-based training is needed.

Adaptive learning involves tailoring learning content, feedback, and interfaces to individual users (Brusilovsky, 2001; De Bra, Aroyo & Cristea, 2004), and the personalization is often automatic. Recently, the effectiveness of adaptive learning has been evaluated in various fields (e.g., Griff & Matter, 2013; Jares et al., 2019; Alwadei et al., 2020). In adaptive learning, memory mechanisms are important elements (Zhou, Nelakurthi & He, 2018; Lalwani & Agrawal, 2019; Zaidi et al., 2020) and have been a longstanding research topic. The forgetting curve hypotheses that Ebbinghaus (1885) proposed are the first theories revealed by experiments in the study of memory mechanisms. The forgetting curve can be explained as follows: the most rapid increase in memory occurs after the first learning, the content learned is exponentially forgotten after learning, repeated learning at intervals (repetition learning and spaced learning) increases the amount of time that memory can be retained, the more information that came in immediately before, the more it is retained in short-term memory, and the more it can be remembered (the recency effect) (Ebbinghaus, 1885; Dempster, 1989; Glenberg, 1979). Learning effectiveness of repetition learning and spaced learning is thought to be improved by setting appropriate learning intervals, as revealed by meta-analyses (Cepeda et al., 2006; Donovan & Radosevich, 1999). The recency effect is that when asked to recall a list of items in any order, people tend to recall from the end of the list and tend to recall those items best (Ebbinghaus, 1885). The memory mechanisms are qualitatively applicable in a quiz-style online training tool on birds through controlling order of bird species in the quiz, but there is no a priori information that quantitatively optimizes the number of quiz training questions, the time from quiz to test, or interval between questions. Understanding how theories relate to training effectiveness may help guide the development of online training tools for learning bird identification from recorded songs and for improving interest in birds and natural environments.

We developed a new online training tool, TORI-TORE, consisting of multiple-choice quizzes for improving bird identification skills from recorded bird songs and fostering citizen scientists skilled in birdsong identification. In this study, we (1) compared species identification skills and attitudes toward birds based on pre- and post- test results and training questionnaires, (2) compared test results and attitudes in a randomized controlled trial to reveal if automatically personalized online training (hereinafter, “adaptive training”) inspired by adaptive learning is more effective than conventional training (hereinafter, “baseline training”), and (3) tested the prediction that a large number of quiz training questions, short lag between training and testing, and long question intervals improve species identification test scores based on forgetting curve hypotheses.

Materials & Methods

Overview of TORI-TORE

We developed a quiz-based birdsong training tool, TORI-TORE, to evaluate the effectiveness of automatically personalized adaptive training. Users can access the tool through a web browser and are identified by cookie at the specified URL (NIES, 2023). Users can access bird sound files stored on the server and choose bird species in a multiple choice quiz. TORI-TORE judges correctness, displays the results on the terminal, and stores correct and user-selected species in addition to quiz choices in the server database (Fig. 1).

Figure 1 User interface of the birdsong training tool TORI-TORE (quiz and answer matching interfaces for the quiz training module in Japanese).

(1) Click to save and log out of the quiz training module. (2) Audio for a bird is automatically played on the question screen. To stop the sound, click “ ||”. (3) Hover the mouse over a choice to change its color and click on the species to select it. (4) Whether the selected species is correct or incorrect is displayed. (5) Users can listen to the sound source for each choice. (6) Users can see an explanation of the bird songs. (7) A photo of the correct choice is displayed. (8) A spectrogram of the correct sound source is displayed. (9) Users can track their progress.

TORI-TORE mainly consists of “test” and “quiz training” modules for evaluation and training, respectively. Quiz training consists of five choices (one correct answer and four distracters) for convenience. Only the correctly selected species will be judged as “correct”, and any other choice will be considered “incorrect”. In the quiz training module, after selecting a species name, the answer screen shows the correct species, the species selected by the user, the correctness, the sound source for each choice, a photo and spectrogram of the correct species, and if the species was “mastered” (described in “Adaptive training algorithm”). In the adaptive group, the history of correct answers affected the next quiz (described in “Adaptive training algorithm”). The goal is to acquire bird songs by repeated listening while taking the quiz training and checking answers. For the test module, see Study Design and Data Collection. There is no time limit for modules in TORI-TORE. TORI-TORE was coded in Perl v5.26.3 and is implemented as a CGI script with jQuery 3.4.1.

Adaptive training algorithm

We developed an adaptive training algorithm that tailors questions based on an individual’s proficiency level, one element of adaptive learning, for efficiently memorizing bird songs. The algorithm was designed (1) to reduce the frequency of bird songs (correct choices) once memorized and (2) to make incorrect alternatives easier when the user was not very proficient in the correct choice and harder when he or she was more proficient.

For creating the adaptive training algorithm, we referred to Tsumori & Kaijiri (2007). To help students memorize vocabulary, Tsumori & Kaijiri (2007) developed an algorithm able to automatically determine questions to be asked depending on the students’ understanding of the vocabulary, using multiple-choice questions in which the level of difficulty was controlled. The specific algorithms were as follows (Fig. 2). All species were given an initial proficiency level of zero. If the user selected the correct choice, the user’s the proficiency level of it increased by one. Selecting an incorrect choice reduced the user’s proficiency levels of the incorrect alternative and the correct choice by one. However, the proficiency level could not decrease below zero. The correct choice was basically selected from species with a proficiency level of less than three; accordingly, the probability of it being included as a correct choice in the training decreased when the proficiency level of the species was more than two (“mastered”). However, mastered species had a certain probability of being included in the training for review. The review probability P is formulated: P=pwNuNu+wNm, where p is max review probability (set at 0.25 in this study), w is review weight (set at 0.5), Nu is number of unmastered species, and Nm is number of mastered species. In the beginning of the training, the correct choices were randomly determined because proficiencies for all species were less than three.

Figure 2 Flowcharts of algorithm (A) to generate answer choices in adaptive and baseline training and (B) to change proficiency levels for each species in adaptive training.

*1 Mastered and unmastered species are described in (B). *2 Pre-determined probability is described in “Adaptive training algorithm” in the main text.*3 Similar species (including easier and more difficult one) is described in “Adaptive training algorithm” in the main text.*4 If there is more than one species with the same proficiency, the algorithm select them randomly.*5 Proficiency level is given to all species and updated every question.

As the proficiency for a bird species gradually changed, alternatives changed accordingly. In other words, the incorrect alternatives were determined by the proficiency level for the correct choices. When the proficiency level of the correct choice was low, species with high proficiency levels were selected as incorrect alternatives. As the proficiency level of the correct choice increased, species with low proficiency levels or “similar” species were selected as incorrect alternatives. “Similar” species were defined as species with similar songs to those of the correct choice, as determined by bird experts. In this study, there are zero to one similar species per correct answer choice (the algorithm allows more than one to exist). The degree of similarity was set to two levels (easier and more difficult level; see Table 1).

Table 1 Target species for training.

ID	English name	Binomial	Abbreviation	Similar species (similarity level)	Song type	
1	Brown-eared Bulbul	Hypsipetes amaurotis	Hyam	–	Call	
2	Japanese Bush Warbler	Cettia diphone	Cedi	–	Song	
3	Eurasian Tree Sparrow	Passer montanus	Pamo	–	Call	
4	Large-billed Crow	Corvus macrorhynchos	Coma	Carrion Crow (E)	Call	
5	Chinese Hwamei	Garrulax canorus	Gaca	Narcissus Flycatcher (E)	Song	
6	Common Pheasant	Phasianus colchicus	Phco	–	Call	
7	Meadow Bunting	Emberiza cioides	Emci	–	Song	
8	Japanese Tit	Parus minor	Pami	Varied Tit (E)	Song	
9	Lesser Cuckoo	Cuculus poliocephalus	Cupo	–	Song	
10	Japanese White-eye	Zosterops japonicus	Zoja	–	Song	
11	Carrion Crow	Corvus corone	Coco	Large-billed Crow (E)	Call	
12	Oriental Greenfinch	Chloris sinica	Chsi	–	Song	
13	Oriental Reed Warbler	Acrocephalus orientalis	Acor	–	Song	
14	Eurasian Skylark	Alauda arvensis	Alar	–	Song	
15	Japanese Wagtail	Motacilla grandis	Mogr	White Wagtail (D)	Song	
16	Barn Swallow	Hirundo rustica	Hiru	–	Call	
17	White Wagtail	Motacilla alba	Moal	Japanese Wagtail (D)	Song	
18	Oriental Turtle Dove	Streptopelia orientalis	Stor	–	Call	
19	Grey Wagtail	Motacilla cinerea	Moci	–	Song	
20	Varied Tit	Poecile varius	Pova	Japanese Tit (E)	Song	
21	Japanese Green Woodpecker	Picus awokera	Piaw	–	Call	
22	Common Cuckoo	Cuculus canorus	Cuca	–	Song	
23	Asian Stubtail	Urosphena squameiceps	Ursq	–	Song	
24	White-cheeked Starling	Spodiopsar cineraceus	Spci	–	Call	
25	Narcissus Flycatcher	Ficedula narcissina	Fina	Chinese Hwamei (E)	Song	
26	Japanese Pygmy Woodpecker	Dendrocopos kizuki	Deki	–	Call	
Notes.

Bird vocalizations are mainly divided into songs and calls. Since data were collected during the breeding season, we used songs for singing birds and calls for non-singing birds. For species with multiple types of songs and calls, the most characteristic vocalization was selected based on expert opinion. Similarity level is set to two levels; (E) is easier and (D) is more difficult level.

Case study

Training target species

The main goal of TORI-TORE is to improve the ability of users to identify the songs of familiar birds in a region of Japan based on recorded data. In a case study to clarify the effectiveness of the efficient quiz-based online training, we focused on acoustic monitoring with IC recorders during the breeding season of birds east of the Abukuma River in Fukushima Prefecture, including the evacuation zone of the Fukushima Daiichi Nuclear Power Plant accident (Fukasawa et al., 2017b). Notably, TORI-TORE was implemented based on experiences from the Bird Data Challenge, a citizen science program conducted in 2015–2018 (Fukasawa et al., 2017a). The Bird Data Challenge is a regional program that citizens identify bird species from a part of environmental sounds recorded by the acoustic monitoring project. The main monitoring target was familiar birds living near human settlements, which are highly sensitive to land abandonment and decontamination of residential areas and agricultural land. The data processed by the Bird Data Challenge was included to the open scientific data set to evaluate the biodiversity status of the evacuation zone (Fukasawa et al., 2017b). The participants of the program were mainly trained bird watchers, and the number of such trained citizens were considerably limited. Given that a large number of recorded sounds has been collected, increase of participants supporting a limited number of trained citizen scientists was expected to facilitate data construction and thus the monitoring project. In the context of whether the participants at the Bird Data Challenge could identify species from recorded birdsong, we selected species to be used for TORI-TORE. In this study, the top 26 species with the highest occurrence rate in the annotated acoustic data, including those obtained from the Bird Data Challenge, were used for training and testing (Table 1). If participants can identify all 26 target species in the training, they can identify most of the species with the monitoring including identifications in the Bird Data Challenge. Vocalization data for training and testing were provided by the Japan Bird Research Association and xeno-canto (Table S1).

Study design and data collection

Participants were recruited in FY 2020. Eighty-four university students (from freshman to senior undergraduates) participated in this study. Participants were from the Kanto region, mainly Ibaraki Prefecture, and had no previously involvement in research or extracurricular activities related to birds. They gave consent for their participation and data use by reviewing a consent letter and checking a consent box.

Participants were assigned by stratified randomization to the adaptive training group (hereinafter, “adaptive group”) or the quiz training group, in which choices were selected at random (hereinafter, “baseline group”). Groups were matched with respect to academic year and gender, and were divided using random numbers.

Participants agreed in advance to the following: (1) they would be randomly assigned to two different quiz training groups, (2) they must not check information about birds on external sites other than the URLs presented in TORI-TORE, (3) they must not discuss the content of the experiment with others, (4) they would receive an honorarium only if they completed the entire experiment.

In the test module, to ensure that the test did not affect the training effect, correct answers were not disclosed to participants until after the delayed test. All 26 target species were included as choices to facilitate evaluation of changes in test scores. The experiment was conducted from January 12 to February 1, 2021. Participants were instructed to conduct the training on the schedule described in Table 2 and Fig. 3.

Table 2 Schedule of experiments.

Schedule	Contents	
Day 1	Pre-training questionnaire, confirmation of audio settings, and pretest	
Day 2 to 3	Training (50 questions each)	
Day 4	Midterm test	
Day 5 to 6	Training (50 questions each)	
Day 7	Posttest and post-training questionnaire	
Day 21	Delayed test	
Notes.

At the commencement of each phase of the experiment, each group received links to TORI-TORE. If they failed to complete the assigned training on a given day, they were not allowed to train on the next day (i.e., they were considered dropouts). There was no time limit to the experiment if it was completed on the appropriate day. The experiment was conducted at participants’ homes using their computers owing to the COVID-19 pandemic.

Figure 3 Scheme of experiment procedure.

The number of valid responses (i.e., participants who completed the experiment) was 66, including 35 (53.0%) males and 31 (47.0%) females. Seventeen each were freshmen (18–20 years old), sophomores (19–21 years old), and seniors (21–23 years old) and 15 were juniors (20–22 years old). A total of 18 participants withdrew from the study.

Questionnaires

We developed web-based structured questionnaires to understand participants’ pre-training experience with nature and attitudes towards birds, to assess the impact of the training, and to improve the tool (Article S1 and Table S2). The questionnaires consisted of pre- and post-training questionnaires and questions were based on studies of the effectiveness of online training in citizen science projects and awareness among conservation activity participants (White, Eberstein & Scott, 2018; Starr et al., 2014; Ratnieks et al., 2016; Fukasawa et al., 2017a; Takase, Furuya & Sakuraba, 2014).

The pre-training questionnaire included 12 items across four sections. Section one was related to attitudes towards birds. On a 5-point Likert scale, participants ranked their interest in birds (1 = “Very interested” to 5 = “Not at all interested”), including birdwatching and learning bird songs. They were also asked to rank their birdwatching and birdsong identification experience level as “No experience”, “Beginner”, “Intermediate”, or “Advanced”. The second section focused on participants’ personal experience with nature, including experience with nature over the past year or environmental activities and pet ownership. The third section focused on motivations to participate in the survey. The final section (sociodemographic status) focused on the participants’ background, specifically their hometown.

The post-training questionnaire consisted of 22 items across two sections. Section one was related to attitudes towards birds. Using a 5-point Likert scale, participants ranked changes in their interests in birds (1 = “My interest in birds has really changed” to 5 = “My interest in birds has not changed at all”) and the level of interest in birds (1 = “Very interested” to 5 = “Not at all interested”), including birdwatching and learning bird songs. On a 5-point Likert scale, participants indicated whether they knew the species name in TORI-TORE and had ever heard the bird songs. The second section focused on the usage of TORI-TORE. We used a 5-point Likert scale to rank satisfaction (1 = “I’m satisfied with this training” to 5 = “I’m not satisfied with this training”) and evaluations (1 = “Strongly agree” to 5 = “Strongly disagree” and 1 = “There were too many questions” to 5 = “There were few questions”). We also used a 5-point Likert scale to assess whether participants perceived the adaptive training algorithm (1 = “Very much” to 5 = “Not at all”) (Article S1 and Table S2). Finally, we asked for feedback as an open-ended question.

Data analysis

To determine whether TORI-TORE was effective, we compared species identification test results and participants’ attitudes based on pre- and post-training questionnaires (see Code S1 and Data S1). Generalized linear mixed models (GLMMs) with a binomial error distribution were used to evaluate whether the training methods (adaptive training and baseline training) affect the posttest score considering the pretest score for each question. As a response variable, a dummy variable was set to 1 if the test was answered correctly and 0 otherwise. As explanatory variables, a dummy variable was set to 1 if the participant had taken the midterm test after adaptive training and 0 otherwise; 1 if the participant had taken the posttest after adaptive training and 0 otherwise; 1 if the participant had taken the delayed test after adaptive training and 0 otherwise; 1 if the participant had taken the midterm test after baseline training and 0 otherwise; 1 if the participant had taken the posttest after baseline training and 0 otherwise; 1 if the participant had taken the delayed test after baseline training and 0 otherwise. The dummy variables were fixed effects, while participants and tested species were included as random slopes and intercepts in the model. In addition, ordered logit models were used to evaluate whether the training method affects the post-training questionnaire responses about interests in birds considering pre-training responses. As a response variable, the questionnaire responses were set, and as explanatory variables, a dummy variable was set to 1 if the participant had taken adaptive training and 0 otherwise and to 1 if the participant had taken baseline training and 0 otherwise.

Second, a GLMM was used to understand whether the test scores after training differed between adaptive training and baseline training groups (see Code S1 and Data S1). The response variable was dummy variables set to 1 for correct answers in each test (midterm test, posttest, or delayed test) and 0 otherwise. The explanatory variable was a dummy variable taking a value of 1 for adaptive training and 0 for baseline training. The dummy variables were fixed effects, while participants and tested species were included as random slopes and intercepts in the model. The Wald tests were conducted to understand whether the posttest scores for each species after training differed between adaptive and baseline groups with the null hypothesis that the difference between the groups was zero. In addition, we used generalized linear models (GLMs) with binomial error distribution or ordered logit models to evaluate whether post-training questionnaire responses differed between groups. GLMs were used when there were two discrete choices, and the ordered logit model was used when there were more than two choices. The response variable was the answer to the questionnaire (five levels: see Article S1 and Table S2), and the explanatory variable was a dummy variable that was set to 1 for adaptive training and 0 for baseline training. To make it easy to interpret the ordered logit models, option numbering was adjusted so that the numbers assigned to positive responses were larger and those assigned to negative responses were smaller. For example, “very interested” was set to 5 and “not at all interested ” was set to 1.

We evaluated the hypothesis that a large number of quiz training questions, short lag times between quiz training and testing (hereinafter “lag times”), and long question intervals improve species identification test (posttest) scores (from the theories of the forgetting curve (Ebbinghaus, 1885)) (see Code S1 and Data S1). First, to understand whether adaptive training affected these factors, we performed MANOVA. We used a dummy variable taking a value of 1 for adaptive training and 0 for baseline training as an explanatory variable, the number of quiz training questions, inverse lag time (/days), and median question interval as the response variables. There was multicollinearity in the number of quiz training questions because the total number of questions in both groups was equal to 200. For this reason, we calculated p-values based on modified ANOVA-type statistics (Friedrich, Konietschke & Pauly, 2021), which can incorporate multicollinearity among response variables. Values were largest when tested immediately after training and were smaller but not equal to zero as time elapsed. We then ran a GLMM with these variables as the explanatory variables and a dummy variable as the response variable, taking a value of 1 for the correct answer in each test and 0 otherwise, to determine whether these variables affected the scores. The explanatory variables were fixed effects, while participants and tested species were included as random slopes and intercepts in the model.

The GLMMs and GLMs were implemented with the glmmTMB function in the glmmTMB package for R version 4.1.0 (Brooks et al., 2017; R Core Team, 2020). The ordered logit models were implemented with the clm function in the ordinal package for R version 4.1.0 (R Core Team, 2020; Christensen, 2019). The MANOVAs were computed using the manova function implemented in R version 4.1.0 (Brooks et al., 2017; R Core Team, 2020). In the case of multicollinearity, the MANOVA.wide function was used in the MANOVA.RM package for R version 4.1.0 (Friedrich, Konietschke & Pauly, 2021; R Core Team, 2020).

Ethics statement

Approval for this study was granted by the University of Tsukuba’s Faculty of Life and Environmental Sciences Ethics Committee (Subject number 2020-1). All tests, training, and questionnaire responses were anonymous. Each participant assigned a unique number as an identifier to match tests and training datas, and questionnaires for a given individual.

Informed consent was obtained from all participants. We explained the following to the participants before the experiment. Participation in the experiment was determined by the participants’ own free will. Therefore, they would not be disadvantaged in any way if they did not agree to take part in this experiment. In addition, even after consenting to participate in the experiment, they may withdraw from participation at any time and would not be disadvantaged by this.

Results

Effects of TORI-TORE

In the midterm test, posttest, and delayed test, scores for both groups were significantly higher than those in the pretest (GLMMs: p-value for the Z-statistic < 0.001, Table S3). Scores for both groups increased from the pretest to midterm test and midterm test to posttest, but decreased from the posttest to delayed test (Fig. 4, GLMMs: p-value based on the Z-statistic < 0.001, Table S3). In particular, scores were 3.65 (±0.21) in the pretest, 11.45 (±0.57) in the midterm test after 2 days of training (100 questions), 14.62 (±0.71) in the posttest after 4 days of training (200 questions), and 12.45 (±0.69) in the delayed test.

Figure 4 Relationship between time elapsed (days) from the pretest and the score in each group.

The midterm test was administered on day 3, the posttest was administered on day 6, and delayed test was administered on day 20. The total score was 26 points. Black asterisks (or dots) indicate a significant difference in scores between the two groups in the midterm to delayed test, and blue (red) asterisks indicate a significant difference in scores in the adaptive group (baseline group) from the pretest to midterm test, midterm test to posttest, and posttest to delayed test. ***p < 0.001, **p < 0.01, *p < 0.05, .p < 0.1.

Both groups showed increased interest in birds, bird watching, and learning bird songs based on pre- and post-training questionnaire responses (ordered logit models: p-value based on the Z-statistic < 0.01). For interest in birds, six participants in the adaptive group (22.2%) and six participants in the baseline group (15.4%) answered “very interested” or “interested” (positively) before the training, compared with 21 (77.8%) and 33 (84.6%) after the training. For interest in bird watching, 10 (37.0%) and nine (23.1%) responded positively before the training, compared with 21 (77.8%) and 33 (84.6%) after the training. For interest in birdsong learning, seven (25.9%) and 18 (46.2%) responded positively before the training, compared with 21 (77.8%) and 33 (84.6%) after the training.

Comparison between adaptive training and baseline training

In the midterm test, the adaptive group scored 9.4 (SE ±0.7) and the baseline group scored 12.8 (SE ±0.8). In the posttest, scores were 12.7 (SE ±1.2) and 15.9 (SE ±0.8). In the delayed test, scores were 10.9 (SE ±1.1) and 13.6 (SE ±0.9) respectively. Scores for the baseline group were significantly higher than those for the adaptive group in all tests (Fig. 4 and GLMMs: p-value based on the Z-statistic = 0.0021, 0.037, 0.073, respectively, Table S4). Baseline training had a more positive influence than that of adaptive training on test scores in the comparison between the pretest and the midterm test with a wider score gap. However, adaptive training had a more positive influence than that of baseline training from the midterm test to posttest and had a less negative impact than baseline training on the change from the posttest to delayed test, with a narrower difference in scores between the two groups (Fig. 4 and GLMMs: p-value based on the Z-statistic < 0.001, Table S4).

In the posttest, there was a variation in the accuracy rate for each species and each group (Fig. 5). There were significant differences in the scores between the groups (Wald test summary in Table S5) for Common Pheasant (Phco, p = 0.033), Grey Wagtail (Moci, p = 0.00040), Japanese Pygmy Woodpecker (Deki, p = 0.0048), Oriental Greenfinch (Chsi, p = 0.030). In the adaptive training algorithm, similar species were included as choices when the proficiency level for a species increased; accordingly, the adaptive group was expected to have a higher accuracy rate for similar species. Contrary to this expectation, the accuracy rate for similar species was not higher in the adaptive group compared with the baseline group, except for large-billed crow (Coma) and carrion crow (Coco).

Figure 5 Relationship between species and accuracy rates in the posttest.

Error bars represent the standard error. Species are arranged on the x-axis according to the mean accuracy rate (from high to low).

There was no difference between groups in questions related to attitudes towards birds on the post-training questionnaire (ordered logit models: p-value based on the Z-statistic > 0.05). Regarding the “usage of TORI-TORE”, responses in both groups were generally positive. Regarding whether participants perceived the adaptive training algorithm, the adaptive group exhibited significantly different responses to the following items: “Wrong species in the quiz were followed by more questions in the quiz”, “The frequency of questions on the mastered species decreased”, “As the proficiency level of the species of the correct choice increases, a similar species is selected as the incorrect alternative (but not at lower proficiency levels)”, “While the species of the correct choice was less proficient, the more proficient species was selected for the incorrect alternatives, and as the proficiency level increased, the less proficient species were selected for these incorrect alternatives” (ordered logit models: p-value based on the Z-statistic < 0.05). In other words, participants of adaptive group recognized that they were receiving adaptive training. Adaptive training had a significantly negative effect on the responses about satisfaction (ordered logit models: estimate = −1.307, p-value based on the Z-statistic = 0.0121). No significant differences between groups were found for other questions (ordered logit models: p-value based on the Z-statistic > 0.05).

Factors that contribute to training effectiveness

Variables affected by training methods

As expected, there were significant differences in the effects of the number of quiz training questions, the inverse lag times, and the median question intervals for each of the 26 species at the time of the posttest between the adaptive and baseline groups (MANOVA: p < 0.001, p = 0.0094, p = 0.066, respectively, Table S6).

Variables affecting test scores

The number of quiz training questions, inverse lag time, and median question interval (explanatory variables), irrespective of group, influenced test scores (response variable) (GLMM: p- value based on the Z-statistic < 0.001, respectively, Table S7). Specifically, a large number of quiz training questions, a long inverse lag time (few lag times), and a long question interval had a positive effect on test scores. In other words, the training method affected these three parameters, which in turn affected test scores.

Relationship between explanatory variables and the accuracy rate for each species

For the relationship between number of quiz training questions and the accuracy rate for each species, the number of quiz training questions tended to be 6–9 for the baseline group and 4–10 for the adaptive group (Fig. 6). In the adaptive group, a higher number of quiz training questions corresponded to species with lower accuracy rates. In addition, for the random slope for the effect of each species (i.e., how much the score improves for each quiz training question) and the number of quiz training questions, the correlation coefficients were −0.223 (p = 0.27) for the baseline group and −0.546 (p = 0.0039) for the adaptive group. For the relationship between inverse lag time and the accuracy rate for each species, the baseline group was concentrated at 0.8–1.0 /days (1–1.25 days), while the adaptive group was concentrated in 0.6–1.1 /days (0.91–1.67 days) (Fig. 7). In the adaptive group, an increased inverse lag time (i.e., to the right of the graph) corresponded to species with lower accuracy rates in the adaptive group. In addition, for the random slope for each species (i.e., how well the response improves with the size of the inverse lag time) and the inverse lag time, the correlation coefficient was −0.0172 (p = 0.93) for the baseline group and −0.237 (p = 0.24) for the adaptive group. For the relationship between median question interval and the accuracy rate for each species, the distribution differed from those for two explanatory variables above (Fig. 8). Compared with the baseline group, the adaptive group tended to have more species with narrower intervals (e.g., the adaptive group had 18 species with values below 20, compared with 13 species for the baseline group). A higher median question interval (i.e., to the right of the graph) corresponded to species with higher accuracy rates and a lower median question interval corresponded to species with the lower accuracy rates in the adaptive group (more than in the baseline group). In addition, for the random slope for each species (i.e., how well the response improves with the size of the median question intervals) and the median question interval, the correlation coefficient for the adaptive group was 0.320 (p = 0.11) and for the baseline group was 0.475 (p = 0.014).

Figure 6 Relationship between number of quiz training question and accuracy rates in the posttest.

Figure 7 Relationship between inverse lag time and accuracy rates in the posttest.

Figure 8 Relationship between median question interval and accuracy rates in the posttest.

For Common Pheasant (Phco), Grey Wagtail (Moci), Japanese Pygmy Woodpecker (Deki), and Oriental Greenfinch (Chsi), where there were significant differences in scores between groups on the posttest, the results in terms of accuracy rate, tendency to answer incorrectly, and each explanatory variable are as follows. Regarding the difficulty level, the mean accuracy is medium for Grey Wagtail (Moci), Japanese Pygmy Woodpecker (Deki), and Oriental Greenfinch (Chsi), but the accuracy of Common Pheasant (Phco) was exceptionally high (Fig. 5). In addition, there is not much difference in the number of quiz training questions for Grey Wagtail (Moci), Japanese Pygmy Woodpecker (Deki), and Oriental Greenfinch (Chsi) between groups, but the number of quiz training questions for the adaptive group of Common Pheasant (Phco) is lower than that for the baseline group (Fig. 6). For Oriental Greenfinch (Chsi), the adaptive group tended to answer more times and at shorter question intervals. By the adaptive group, Oriental Greenfinch (Chsi) tended to be misidentified to species with larger number of quiz training questions and narrower intervals. Grey Wagtail (Moci) tended to be misidentified as other wagtails (Japanese Wagtail (Mogr) and White Wagtail (Moal)) by the adaptive group, and Japanese Wagtail (Mogr) and White Wagtail (Moal) were more likely to be listed as a similar species in the same choice in the adaptive training (Table 1, “Adaptive training algorithm”). However, the song of Grey Wagtail (Moci) is not similar to that of Japanese Wagtail (Mogr) and White Wagtail (Moal), and was not listed as a similar species. In addition, the adaptive group had narrower question intervals than the baseline group for Oriental Greenfinch (Chsi) and Grey Wagtail (Moci) (Fig. 8). For Japanese Pygmy Woodpecker (Deki), however, no trend was observed in the number of questions submitted and the intervals by themselves, or the species misidentified.

Discussion

TORI-TORE, a newly developed quiz-style online training tool, improved birdsong identification and interest in birds, providing a basis for manual identification of bird species in acoustic monitoring datasets. Based on mean values, participants (university students with no experience in bird watching-related activities) were able to identify more than half of the species after 4 days of training. Two weeks after training, there was a slight drop in scores from the posttest scores. Regarding interest in birds, bird watching, and learning bird songs, the percentages of positive responses increased in both groups, with more than 60% of respondents giving positive responses. These results are consistent with previous research indicating that video-based online training in citizen science increases accuracy in species identification (Starr et al., 2014; Ratnieks et al., 2016) and that direct training improves participants’ attitudes (Hsu, Chang & Liu, 2019). Our findings support the effectiveness of quiz-based online training on both species identification accuracy and attitudes, even for individuals who participated for rewards.

In brief, the adaptive training algorithm determines species proficiency based on the participant’s history of quiz responses and focuses on species with low proficiency or changes the incorrect alternatives according to the proficiency of the species with the correct choice. Although the algorithm was recognizable by participants, the participants who had the algorithm did not outperform those with baseline training (Fig. 4 and Table S8). The baseline group had higher accuracy rates for most species. For similar species, the baseline group had higher accuracy rates, except for crows. It would be expected for the baseline group to have higher accuracy because the adaptive group is facing more difficult questions. In both groups, the difference in scores narrowed from the midterm to the posttest (Fig. 4 and Table S3). The baseline group was subjected to easier quizzes, which may be less efficient than adaptive training, which focuses on harder questions (including similar species). In other words, a longer period of adaptive training may result in smaller differences in scores, including those for similar species. In addition, the adaptive group had a lower satisfaction level; however, this does not necessarily reflect only the difference in the tool.

Although the effectiveness of the adaptive training over the baseline training differed among species (Fig. 3, Table S5), we found no clear relationship with ecological, phylogenetic, geographic, or song characteristics due to small number of species. Because many of the participants in this study had no prior birdwatching experience and limited knowledge of birds, perhaps the characteristics of the songs have affected effectiveness of adaptive training in a complex way. For example, cognitive system overload in remembering similar sounds (Baddeley, 1968) can cause massed learning in adaptive training to not work effectively. Also, difficulty in the long-term retention of monotonous sounds (Ellis & Turk-Browne, 2019) may lead to a pattern of not being able to recall “mastered” sounds during training in tests. It is difficult to statistically verify the effect of voice characteristics on experimental results due to the small number of species in this study. In the future, it would be desirable to optimize the algorithm based on the characteristics of the song by designing experiments that handle a wide variety of species.

To improve the efficiency of birdsong identification by adaptive training, it is necessary to optimize the allocation of effort for each species according to the likelihood of acquisition. Although the adaptive group was subjected to more vigorous training for species with a lower accuracy rate, this approach was not effective (Figs. 5 and 6). Contrary to the theories of the repetition learning (Ebbinghaus, 1885), which states that more training events result in higher accuracy rates, an increase in training questions did not improve scores in the adaptive group (Fig. 6). It is possible that questions in the adaptive group were too biased toward species with low learning efficiency. The median question intervals in the adaptive group were generally narrower than those in the baseline training group (Fig. 8), contrary to the theory of spaced learning, which predicts that solving at wider intervals increases the learning effect (Glenberg, 1979). The adaptive training algorithm can be improved by changing the definition of “mastered”. The participants in this study had no experience in bird watching-related activities, and criterion for mastery may be an inappropriate, that is the algorithm could increase the difficulty at a faster rate than their skills improve. Our results suggested that a certain number of quiz training questions and steady memorization may lead to retention, even for species with a relatively high accuracy rate and considered “easy” by experts. An analysis of lag times suggested that training on as many species as possible just before the test is sufficient but does not necessarily lead to retention. It is necessary to review information at appropriate intervals. The results for the median question intervals for each species suggest that it is necessary to develop an algorithm that adjusts the question interval appropriately such that trainees are not forced to solve difficult questions consecutively over a short period, as in the adaptive group. Because this was a short-term experiment, participants were not able to freely set the training period and the number of quiz training questions. Accordingly, the adaptive training algorithm was not able to properly determine mastery or to effectively personalize the training. This may have led to a “low interval hell”, where the number of questions and intervals were not appropriate for each species (Anki, 2021).

As one of the limitations of this study, the participants were motivated by incentives, rather than interest in birds. It is possible that the results would differ if participants were more interested in birds. The more motivated a person is, the better he or she can remember (Anderman & Dowson, 2011), but since rewards did not vary with performance in this study, we do not consider that rewards will have the effect of increasing performance beyond what is necessary. In addition, the participants were randomly divided into groups that were matched in terms of grade and gender; however, it is possible that there was unexpected bias between the two groups. Furthermore, there was only one sound source for the tests and training. Another limitation of our experimental design is that we did not have a control group which did not use a training program. However, we considered that the pretest results of the participants represented the condition of not using a training program and that the pre- and post-training results should reflect the effects of training because it is unlikely that the scores were improved in such a short time without training. Lastly, the software solution for the smartphone platform is necessary for those who do not have a PC or want to use it on the go, but that is a future issue.

Conclusions

We developed a quiz-based online training tool TORI-TORE, and mainly found that TORI-TORE effectively improves birdsong identification skills and interest in birds compared to before training. Although the adaptive training was not optimized, the approach can be further expanded and refined, including the adjustment of the training duration, number of quiz training questions, and question intervals. Whether it can be used in actual species identification from recorded birdsong is a future task, our tool is expected to improve not only the reliability of acoustic identification data in citizen science projects, but also beyond acoustic identification. Such online training could be broadly applied with implications for training aimed at knowledge sharing (Target19), which is one of the goals of the Aichi Targets (Convention on Biological Diversity, 2010) and in the post-2020 targets (Convention on Biological Diversity, 2020).

Although the dataset used in this research is based on bird songs frequently listened to at monitoring sites in Fukushima, it differs little from the species frequently listened to in many parts of Japan. Therefore, it reflects the basic bird fauna in Japan and could be used in many regions in Japan. However, species not covered by this study’s data set include birds from Hokkaido and the Nansei Islands, seabirds, and alpine birds. It is expected that by modifying the sound dataset to suit the regional bird fauna and by changing the language, it can be used in other regions besides Japan. In conclusion, this research is expected to contribute to capacity building and interest in birds and the natural environment in an increasingly online environment.

Supplemental Information

Article S1 Original data of pre and postquestionnaire (in Japanese)

Click here for additional data file.

Table S1 Table of sound sources used in this study

We trimmed some of the audio data.

Click here for additional data file.

Table S2 Pre- and post-training questionnaires (English)

Click here for additional data file.

Table S3 Summary of the GLMM of the effects of the training methods (adaptive training and baseline training) on test scores

midterm test: effect of taking the midterm test compared with not taking itposttest: the effect of taking the posttest compared with not taking it delayed test: effect of taking the delayed test compared with not taking it

***p < 0.001

Click here for additional data file.

Table S4 Summary of the GLMM of the effect of adaptive training on test scores

Results for the delayed test were marginally significant. p < 0.1; *p < 0.05; **p < 0.01

Click here for additional data file.

Table S5 Summary of the Wald test for differences in posttest scores between groups

.p < 0.1; *p < 0.05; **p < 0.01; ***p < 0.001

Click here for additional data file.

Table S6 Summary of MANOVA of the effect of adaptive training on three variables

Results for the median question interval were marginally significant. p < 0.1; **p < 0.01; ***p < 0.001

Click here for additional data file.

Table S7 Summary of the GLMM of the effect of each variable on posttest scores

*p < 0.05; ***p < 0 .001

Click here for additional data file.

Code S1 The R language code for statistical analyses

Click here for additional data file.

Data S1 Dataset

Click here for additional data file.

We thank our colleagues and many participants for operation checks and surveys of TORI-TORE. We also thank the Bird Research Association for providing sound sources of birdsong. We also appreciate the valuable and productive comments provided by the handling editor, Dr. Kush Shrivastava, and the two reviewers (one anonymous and Dr. Ivan Petrushin).

Additional Information and Declarations

Competing Interests

Author Contributions

Human Ethics

Data Availability

The authors declare there are no competing interests.

Yui Ogawa conceived and designed the experiments, performed the experiments, analyzed the data, prepared figures and/or tables, authored or reviewed drafts of the article, and approved the final draft.

Keita Fukasawa conceived and designed the experiments, analyzed the data, authored or reviewed drafts of the article, and approved the final draft.

Akira Yoshioka conceived and designed the experiments, analyzed the data, authored or reviewed drafts of the article, and approved the final draft.

Nao Kumada conceived and designed the experiments, authored or reviewed drafts of the article, and approved the final draft.

Akio Takenaka conceived and designed the experiments, authored or reviewed drafts of the article, and approved the final draft.

Taiichi Ito conceived and designed the experiments, authored or reviewed drafts of the article, and approved the final draft.

The following information was supplied relating to ethical approvals (i.e., approving body and any reference numbers): the University of Tsukuba’s Faculty of Life and Environmental Sciences Ethics Committee granted Ethical approval to conduct the experiment.

The following information was supplied regarding data availability:

Sound sources used in this study, questionnaires, data and code are available in the Supplementary Files.

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
