# Peer review of "Quiz-style online training tool helps to learn birdsong identification and support citizen science"

_PeerJ, doi:10.7717/peerj.15387_

## Round 0.1 · original submission · Minor Revisions

Dear Authors,

Thank you for your submission to PeerJ, I am having reports from reviewers for your manuscript, which, generally is positive. However, reviewers have suggested some minor and important corrections/ suggestions that need to be answered first.

As I was going through your manuscript, I found that its overall presentation and language part is good. The construction of the paper, discussion and introduction are optimum.

However, I agree with the points raised by Reviewer 1 and 2. I don't find the URL in the manuscript to see the webpage of TORI TORE. There was no control ro comparison groups hence you cannot compare the efficiency of TORI TORE against other similar software (which I think was not part of your study too), but having a control group could have been better as per overall testing is concerned.

Usefulness of the tool in question should be very briefly discussed as stated by reviewer 2.

As indicated by reviewer 1, and also mentioned by the authors, the baseline group had easier questions. Would this not incorporate a bias in experiment ?
As indicated by reviewer 2, please provide justification for statistical significance.

Please also check minor corrections from reviewer 1.

Hence, it is suggested that you should go through the comments of both the reviewers & make necessary changes/ modifications in the manuscript. Whether the corrections are incorporated or not, it is necessary to give a proper justification for both situations.

Looking forward for your revised manuscript.

Regards,

Reviewer 1 ·

Basic reporting

I would like to understand the bigger picture motivation of this project. There was some discussion of some acoustic survey efforts in the Fukushima Prefecture, but the connection to the project was not entirely clear. How will the bird identification knowledge of these participants contribute to citizen science? The citizen science project iNaturalist relies on volunteers to confirm the identification reports made by other volunteers; the eBird project relies on volunteer bird identifications (visual and auditory); the BirdNET App project allows users to confirm/reject machine-learning based bird sound identifications.

Experimental design

The presentation of adaptive learning “memory mechanisms” should be expanded (68-77). There was very little explanation for the theories that were presented, nor was a clear connection made between the theories, nor was their application to TORI-TORE clear.

Would it be expected for the baseline group to have higher accuracy because the adaptive group is facing more difficult questions? If the algorithm increases the difficulty at a faster rate than their skills improve, I would expect to see a difference in performance. The authors note this in lines 353-358, but I think it is an important note that should be emphasized.

Validity of the findings

It seems that the authors’ study design would not provide a very powerful test of whether TORI-TORE was effective. The authors could evaluate participants attitudes and test scores, but there was no control group that used an entirely different training program or no training program. Thus, it seems that the authors could not evaluate whether TORI-TORE was better than any other program. Therefore, it will be important to carefully define “effective” (eg, we “confirmed the effectiveness of it” (l. 393))

Were there any ecological, phylogenetic, or demographic patterns in which species were easy or difficult to identify? It would be interesting to see additional interpretations of Figure 3. Additional discussion of why some species (Chsi, Moci, Deki) were highly divergent for adaptive vs baseline would be interesting.

Additional comments

Minor Comments
18: approach to what?
29: should be a colon after “results”, not a period
52: “characterized by low costs and accessibility” implies that both costs AND accessibility were low, but I don’t think that’s what the authors mean.
82: this is a prediction, not a hypothesis
137: is it possible that the need to complete the entire experiment in order to receive compensation influenced participants’ experience? Update: the authors partially address this idea in line 386.

·

Basic reporting

According to the manuscript “Users can access the tool through a web browser and are identified by entering a username and password at the specified URL.”, but no URL is provided to use the tool; Some CGI script was mentioned “TORI-TORE was coded in Perl v5.26.3 and is implemented as a CGI script with jQuery 3.4.1.”, but source not available. It’s better to publish the source of web application as GitHub repository.
The mentioned appendix 1 is missing.
As mobile application developer I suggest to adapt the software solution for the smartphone platform. The user interface of application at Figure 1 looks like PC-based version, but it’s hard to use PC outdoors.
Speaking of style, I don't feel qualified to judge about the English language and style, but most sentences are easy to understand.

Experimental design

The sound dataset of bird songs is missing, one cannot reproduce the results of the study with different participants group. The questioning procedure is described fine, but better be supported with scheme.

Validity of the findings

The confidence intervals for several species at Figure 3 are intersecting for adaptive and baseline groups. It shows that statistically significant difference is questionable and requires accurate statistical evaluation of the data.
Discussion section is brief and contain the comparison of effectiveness of TORI-TORI with analogous tools. How this tool can be used in other locations of Japan? Please suggest the application of the tool in the Conclusion.

---

## Round 0.2 · Minor Revisions

Dear Authors,

Thank you for submitting your revised manuscript to PeerJ. I have received reviewer reports for your revised manuscript. In general, the reports are positive for your manuscript. However, both reviewers have suggested minor revisions. As I was going through the manuscript, it seems that with a little more effort, the manuscript can attract a wider range of audience. hence, I would like to request the authors put some more effort to make this manuscript more far-reaching and interesting for the broader category of researchers.

Kindly go through the comments of each reviewer and prepare your revised manuscript/ response accordingly. Should you choose not to include the comments kindly give a logical rebuttal to that comment. As, I was going through your revised manuscript I found that the revised portion is having few grammatical mistakes (for eg. line 71-72) etc. Also, there is some need for improvement in sentence construction/ English language part, to make the points clear and understandable. Please check for these in the revised version.

Looking forward to your revised manuscript,

Regards

Reviewer 1 ·

Basic reporting

There is some odd formatting throughout that makes it difficult to discern the paragraph breaks. It seems that some paragraphs are extremely short (one sentence).

I suggest reading the paper ‘The machine learning–powered BirdNET App reduces barriers to global bird research by enabling citizen science participation’ in PLOS-Biology. It seems relevant to the introduction, as it describes a citizen science tool that uses sound to identify birds (disclosure: I am one of the authors)
It would be valuable to know how the “distracter” choices (ie, the four incorrect choices in the quizzes) were selected and whether there were patterns in incorrect answers. Were the “distracters” randomly generated, or were there songs of species that are particularly similar to the correct species (and if so, how many?). This could be a critical component of how difficult a question is, and it was not clear to me whether or how this was accounted for. There is one sentence about this in lines 312-313 (“In the adaptive training algorithm (Article S1), similar species were included as choices when the proficiency level for a species increased; accordingly, the adaptive group was expected to have a higher accuracy rate for similar species”). Later in the new text in the Discussion (lines 399-430), the authors mention some patterns in the incorrect guesses. However, I think that kind of information is really important and should be in the main text, not the supplementary material.

After reading the Article S1, I think it should be included in the text. It is not particularly technical and it is not very long. However, it was a little bit difficult to follow, so a diagram could help the reader understand. The adaptive learning algorithm seems like a key part of the author’s study, so it would be worth including a thorough explanation.

The authors added a lot of new text to the Discussion (lines 399-430). Much of it was inappropriate for that section and I do not think it has made the manuscript better. There was essentially a long list of species-specific results – which should go in the results section, especially because it was very dense information and I started to lose track of what I was reading. There was a little bit of higher-level summary text in there, and that is what belongs in the Discussion. The key patterns, not a huge block of results.

Experimental design

na

Validity of the findings

na

Additional comments

Minor Comments (track changes version, “simple” view)
71: “hypotheses”
84: “prior”
89: “fostering citizen scientists of birdsong identification” doesn’t quite make sense. Maybe a word is missing…
97-100: the text “In adaptive training…baseline training” seems like Methods content that does not belong at the end of the introduction
309 and many other places: Most journals do not want you to state “our results are in Table S9”. Instead, if the results are important, they should be described and the table is referenced parenthetically.
375: “basis for manual identification of bird species in acoustic monitoring datasets”. This is an important point, because there are machine learning tools that can automatically identify birdsong (probably more efficiently than volunteers)
389: The authors implicitly state that the algorithm did not outperform baseline training. Yet as I understand the paper, really it was participants who had the algorithm that didn’t outperform participants with baseline training.
444: “the algorithm could have been increasing the difficulty…”. I agree – I was very surprised that just three correct answers was considered “mastered”, as that seems very low.

·

Basic reporting

Thank you for trial access to the web tool. When viewing on mobile device I see an errors about browser sometimes. The website layout should have mobile version to see all the details of the interface. I think most of the users will access the web tool via iOS or Android OS.

Experimental design

The changes in manuscript and supplementary are thorough. We can discuss the development of Android app for this task in future, my students take the class of mobile app development. Please contact [email protected] if interested.

Validity of the findings

I suggest to rephrase the title to avoid word “effectiveness” and it shorter, f.e.: “Quiz-style online training tool helps to learn birdsong identification and support citizen science”. The control group is required to evaluate the effectiveness.
It’s hard to find the similar software or studies to compare with, I understand. But there are two papers on ornithology was published recently about birdsong recording and citizen science:
Jäckel, D., Mortega, K.G., Darwin, S. et al. Community engagement and data quality: best practices and lessons learned from a citizen science project on birdsong. J Ornithol 164, 233–244 (2023). https://doi.org/10.1007/s10336-022-02018-8
DIS '20: Proceedings of the 2020 ACM Designing Interactive Systems Conference July 2020 Pages 1687–1700 https://doi.org/10.1145/3357236.3395478

---

## Round 0.3 · Minor Revisions

Dear Authors,

Thank you for submitting your revised manuscript to PeerJ. Your revised manuscript is now better readable and understandable. The report from the reviewer is also positive. There are very few last drafting corrections pointed out by the reviewer that I think are important before final acceptance. Kindly go through the reviewer's comments and prepare your manuscript and response accordingly. it is once again advised that kindly thoroughly check your final manuscript for grammatical and spelling errors, before final submission.
Looking forward to your revised manuscript.

Thanks and Regards,
KS

Reviewer 1 ·

Basic reporting

I think the authors have substantially improved their manuscript. I had previously suggested moving a long section of information about specific bird species to the results; the authors chose to delete it and write a more high-level synthesis paragraph. I think that is a good change, but I encourage the authors to think about those species-specific results! There could be some really interesting patterns to be explored in the future about whether and how some species see easier to learn (or easier to confuse with each other) than others.

Minor comments (line numbers refer to the clean (pdf) version):
96-97: a hypothesis is evaluated, and researchers should be able to make specific predictions about possible outcomes of their hypothesis. Thus, it is somewhat inaccurate to say "[we] evaluated the predication that a large number of quiz training questions..."
125-126: slight wording adjustment needed here, as "choices" are not skilled, people are. Harder choices may require more skill to answer though.
132-133: there is some confusion here. First "proficiency" refers to the user...then it refers to the questions. It cannot be both, and more logically, it is a property of a person. The proficiency REQUIRED to select/avoid a given quiz answer makes sense though.
429: knowledge of birds

Font size in Figure 2 may need to increase to ensure good reproducibility in the journal.

Experimental design

NA

Validity of the findings

NA

Additional comments

NA

---

## Round 0.4 · accepted · Accept

Dear Authors,

Thank you for submitting your revised manuscript to PeerJ. I am happy to inform you that the reviewers have now recommended for acceptance of your article. However, there is a typographical error in line no 402 - 403 as reviewer 1 has identified. Hence, it is recommended to correct it during the publication process.

Thank you for choosing PeerJ for your research publication.

Best Regards,

Reviewer 1 ·

Basic reporting

I found one minor typo (below) but otherwise I have no further comments. The authors have done a good job improving the manuscript over many rounds of revisions.
Line numbers refer to the track changes manuscript - 402-403: there is a typo in "...Common Pheasant (Phco), while for it is higher accuracy."

Experimental design

na

Validity of the findings

na

Additional comments

na